# Tumor necrosis as a predictor of early tumor recurrence after resection in patients with hepatoma

**Yi-Hao Yen**[1⊛‡*], **Fang-Ying Kuo**[2‡], **Hock-Liew Eng**[2], **Yueh-Wei Liu**[3], **Chee-Chien Yong**[3], **Wei-Feng Li**[3], **Chih-Chi Wang**[3⊛*], **Chih-Yun Lin**[4]

**1** Division of Hepatogastroenterology, Department of Internal Medicine, Kaohsiung Chang Gung Memorial Hospital and Chang Gung University College of Medicine, Kaohsiung, Taiwan, **2** Department of Pathology, Kaohsiung Chang Gung Memorial Hospital and Chang Gung University College of Medicine, Kaohsiung, Taiwan, **3** Liver Transplantation Center and Department of Surgery, Kaohsiung Chang Gung Memorial Hospital, Kaohsiung, Taiwan, **4** Biostatistics Center of Kaohsiung Chang Gung Memorial Hospital, Kaohsiung, Taiwan

⊛ These authors contributed equally to this work.
‡ YHY and FYK also contributed equally to this work.
* cassellyen@yahoo.com.tw (YHY); ufel4996@ms26.hinet.net (CCW)

## Abstract

### Background

Tumor necrosis is a significant risk factor affecting patients' prognosis after liver resection (LR) for hepatocellular carcinoma (HCC). We aimed to develop a model with tumor necrosis as a variable to predict early tumor recurrence in HCC patients undergoing LR.

### Materials and methods

Patients who underwent LR between 2010 and 2018 for newly diagnosed HCC but did not receive neoadjuvant therapy were enrolled in this retrospective study. Six predictive factors based on pathological features—tumor size > 5 cm, multiple tumors, high-grade tumor differentiation, tumor necrosis, microvascular invasion, and cirrhosis—were chosen *a priori* based on clinical relevance to construct a multivariate logistic regression model. The variables were always retained in the model. The impact of each variable on early tumor recurrence within one year of LR was estimated and visualized using a nomogram. The nomogram's performance was evaluated using calibration plots with bootstrapping.

### Results

Early tumor recurrence was observed in 161 (21.3%) patients. The concordance index of the proposed nomogram was 0.722. The calibration plots showed good agreement between nomogram predictions and actual observations of early recurrence.

### Conclusion

We developed a nomogram incorporating tumor necrosis to predict early recurrence of HCC after LR. Its predictive accuracy is satisfactory.

**Data Availability Statement:** All relevant data are within the manuscript and its Supporting Information files.

**Funding:** This study was supported by Grant CMRPG8L0181 from the Chang Gung Memorial Hospital-Kaohsiung Medical Center, Taiwan. The funders had no role in study design, data collection and analysis, decision to publish, or preparation of the manuscript.

**Competing interests:** The authors have declared that no competing interests exist.

## Introduction

Hepatocellular carcinoma (HCC) is a leading cause of cancer-related death worldwide [1, 2]. Liver resection (LR) is a curative treatment for patients with HCC [3–6], but the tumor recurrence rate is high after the procedure [7]. Numerous studies have reported predictive models associated with tumor recurrence after LR. The predictors in these models can be categorized into those pertaining to the tumor burden, i.e., tumor size, number, and volume; biological tumor aggressiveness, i.e., alpha-fetoprotein (AFP) level and poor pathological tumor characteristics, such as poorly differentiated tumors, microvascular invasion (MVI), and satellite nodules; and liver function reserve [8–14].

Tumor necrosis is a well-known indicator of poor prognosis of patients with colorectal cancer [15], pancreatic cancer [16], or renal cell carcinoma [17], who underwent surgical resection. Multiple studies have demonstrated that tumor necrosis is an indicator of poor prognosis of patients undergoing LR for HCC [18–23]. To the best of our knowledge, only one previous study has used a model including tumor necrosis as a factor to predict tumor recurrence after LR in patients with a solitary HCC tumor of ≤ 3.0 cm [19]. However, whether this model could be extrapolated to patients with a tumor burden greater than a solitary HCC of ≤ 3.0 cm is unclear.

Therefore, in this retrospective study, we sought to develop a model incorporating tumor necrosis as a variable to predict early recurrence of HCC after LR.

## Materials and methods

The Institutional Review Board of Kaohsiung Chang Gung Memorial Hospital approved this study (reference number: 202201189B0) and waived the need for informed consent due to the retrospective and observational nature of the study design. Between August 8, 2022 and August 7, 2023, the data were accessed for research purposes. The authors had no access to information that could identify individual participants during or after data collection.

Data were extracted from the Kaohsiung Chang Gung Memorial Hospital HCC registry database. The American Joint Committee on Cancer (AJCC)/Union for International Cancer Control (UICC) tumor–node–metastasis staging criteria (the 7th version) [24] were applied to our HCC registry data from July 2010 to March 2018. Within this period, 2820 patients with newly diagnosed HCC were identified, including 862 patients who underwent LR. We excluded patients who were treated with neoadjuvant therapies (n = 56) and those who underwent noncurative LR (n = 50). Curative LR was defined as the complete resection of macroscopic tumors with microscopically negative surgical margins. Ultimately, 756 patients were enrolled in this study. None of the patients enrolled in this study underwent adjuvant therapy.

In our institution, the general principle of surveillance after LR is based on the National Comprehensive Cancer Network guideline recommendations [3]. The diagnosis of tumor recurrence was based on international guidelines [4–6].

### Pathological examination

Pathological examinations were performed by two senior liver pathologists (HLE and FYK). For patients with multiple tumors, tumor necrosis was defined based on the tumor with the most severe necrosis. An Ishak score of 5 or 6 indicated cirrhosis [25]. Tumor grade, i.e., tumor differentiation, was assessed using Edmondson and Steiner's classification [26]. Major resection was defined as the removal of ≥ 3 Couinaud segments. Early recurrence was defined as recurrence within 12 months of LR.

## Statistical analysis

Categorical variables are presented as number and percentage and were compared using the chi-square test. A multivariate logistic regression model was constructed with tumor size ($> 5.0$ cm vs. $\leq 5.0$ cm), tumor number (multiple vs. solitary), MVI (yes vs. no), tumor necrosis (yes vs. no), cirrhosis (yes vs. no), and tumor grade (high grade [grade 3 or 4] vs. low grade [grade 1 or 2]). These variables were chosen *a priori* based on their clinical relevance [8–14, 18–23] and were always retained in the multivariate logistic regression model. The impact of each variable on early recurrence within one year of LR was estimated and visualized using a nomogram. A calibration plot was created to assess the match between the model-predicted probability and the observed probability of early recurrence within one year. Receiver operating characteristic (ROC) analysis was performed to estimate the predictive performance of the proposed model. The concordance index (c-index) and bootstrap (150 resamples) bias-corrected c-index were computed to validate the predictive performance of the model. Furthermore, the predictive performance of the proposed model and of the AJCC/UICC system [24] was compared with the ROC analysis, and the c-index of the two predictive models was compared using DeLong's test. All *p*-values were two-tailed, and a *p*-value $< .05$ was considered statistically significant. All statistical analyses were performed using the computing environment R 4.1.2 (R Core Team, 2020).

## Results

The background characteristics of the patients are shown in Table 1. In the cohort of 756 patients, 250 (33.1%) were $> 65$ years old, 587 (77.6%) were male, 141 (18.7%) had an AFP level of $> 400$ ng/ml, 361 (47.8%) underwent major resection, 402 (53.2%) were positive for the hepatitis B surface antigen (HBsAg), and 252 (33.3%) were positive for the hepatitis C virus (HCV) antibody. On pathological examination, 190 (25.1%) patients had tumors $> 5.0$ cm in size, 111 (14.7%) had multiple tumors, 418 (55.3%) had MVI, 316 (41.8%) had cirrhosis, 32 (4.2%) showed high-grade tumor differentiation, 285 (37.7%) had T1 tumors, 375 (49.6%) had T2 tumors, 35 (4.6%) had T3a tumors, 28 (3.7%) had T3b tumors, and 33 (4.4%) had T4 tumors.

The median follow-up time of the cohort was 3.25 years (interquartile range [IQR]: 0.95–5.0). Tumor recurrence within one year of LR was observed in 161 (21.3%) patients and recurrence after one year was observed in 126 (16.7%) patients. The median overall survival (OS) of patients showing tumor recurrence within one year of LR was 2.274 (IQR: 1.927–3.521) years and that of patients without recurrence within one year was 6.051 (IQR: 5.711–6.390) years. The proportion of patients with a tumor size of $> 5.0$ cm ($p < 0.001$), multiple tumors ($p < 0.001$), MVI ($p < 0.001$), cirrhosis ($p = 0.026$), high-grade tumor differentiation ($p = 0.026$), tumor necrosis ($p < 0.001$), an AFP level of $> 400$ ng/ml ($p < 0.001$), major resection ($p = 0.007$), and a pathology of T3–4 ($p < 0.001$) was higher in the group with tumor recurrence within one year than in the group without recurrence within one year. However, there were no significant differences in age, sex, HBsAg positivity, and anti-HCV positivity between the two groups.

Table 2 shows the estimated regression coefficients of the multivariate logistic regression analysis of the final model to predict tumor recurrence within one year of LR.

Six predictive factors, tumor grade (grade 3 or 4 vs. 1 or 2), tumor size ($> 5.0$ cm vs. $\leq 5.0$ cm), tumor number (multiple vs. solitary), cirrhosis (yes vs. no), MVI (yes vs. no), and tumor necrosis (yes vs. no), were included in the final model for constructing the nomogram (Fig 1). The total number of points on the scale could be obtained by summing the points from for each predictive factor, and a straight line could be drawn down to the point scales. The

**Table 1. Characteristics of patients stratified by early recurrence of hepatocellular carcinoma within one year of liver resection.**

| Variables | Total, n = 756 | Early recurrence within 1 year, n = 161 | Without early recurrence within 1 year, n = 595 | p |
|---|---|---|---|---|
| Age (years) | | | | 0.815 |
| ≦65 | 506 (66.9%) | 109 (67.7%) | 397 (66.7%) | |
| >65 | 250 (33.1%) | 52 (32.3%) | 198 (33.3%) | |
| Sex | | | | 0.393 |
| Men | 587 (77.6%) | 121(75.2%) | 466 (78.3%) | |
| Women | 169 (22.4%) | 40 (24.8%) | 129 (21.7%) | |
| AFP (ng/ml) | | | | <0.001 |
| ≦400 | 614 (81.2%) | 110 (68.3%) | 504 (84.7%) | |
| >400 | 141(18.7%) | 51(31.7%) | 90 (15.1%) | |
| Not available | 1 (0.1%) | 0 | 1 (0.2%) | |
| Resection type | | | | 0.007 |
| Major | 361 (47.8%) | 92 (51.7%) | 269 (45.2%) | |
| Minor | 395 (52.2%) | 69 (42.9%) | 326 (54.8%) | |
| HBsAg | | | | 1.00 |
| Positive | 402 (53.2%) | 86 (53.4%) | 277 (46.6%) | |
| Negative | 352 (46.6%) | 75 (46.6%) | 316 (53.1%) | |
| Not available | 2 (0.3%) | 0 | 2 (0.3%) | |
| Anti-HCV | | | | 0.070 |
| Positive | 252 (33.3%) | 65 (40.4%) | 187 (31.4%) | |
| Negative | 503 (66.5%) | 96 (59.6%) | 407 (68.4%) | |
| Not available | 1 (0.1%) | 0 | 1 (0.2%) | |
| Variables based on pathological examination: | | | | |
| Tumor size (cm) | | | | <0.001 |
| ≦5.0 | 566 (74.9%) | 100 (62.1%) | 466 (78.3%) | |
| >5.0 | 190 (25.1%) | 61(37.9%) | 129 (21.7%) | |
| Tumor number | | | | <0.001 |
| Single | 645 (85.3%) | 114 (70.8%) | 531 (89.2%) | |
| Multiple | 111 (14.7%) | 47 (29.2%) | 64 (10.8%) | |
| Microvascular invasion | | | | <0.001 |
| Negative | 338 (44.7%) | 38 (23.6%) | 300 (50.4%) | |
| Positive | 418 (55.3%) | 123 (76.4%) | 295 (49.6%) | |
| Cirrhosis | | | | 0.026 |
| Presence | 316 (41.8%) | 82 (50.9%) | 234 (39.3%) | |
| Absence | 437 (57.8%) | 79 (49.1%) | 358 (60.2%) | |
| Not available | 3 (0.4%) | 0 | 3 (0.5%) | |
| Tumor differentiation (Edmondson-Steiner grade) | | | | 0.022 |
| Garde 1 or 2 | 719 (95.1%) | 148 (91.9%) | 571 (96.0%) | |
| Grade 3 or 4 | 32 (4.2%) | 13 (8.1%) | 19 (3.2%) | |
| Not available | 5 (0.7%) | 0 | 5 (0.8%) | |
| Tumor necrosis | | | | <0.001 |
| No | | 76 (47.2%) | 401 (67.4%) | |
| Yes | | 85 (52.8%) | 194 (32.6%) | |
| T stage | | | | <0.001 |
| 1 | 285 (37.7%) | 27 (16.8%) | 258 (43.4%) | |
| 2 | 375 (49.6%) | 85 (52.8%) | 290 (48.7%) | |
| 3a | 35 (4.6%) | 17 (10.6%) | 18 (3.0%) | |

(*Continued*)

**Table 1.** (Continued）

| Variables | Total, n = 756 | Early recurrence within 1 year, n = 161 | Without early recurrence within 1 year, n = 595 | p |
|---|---|---|---|---|
| 3b | 28 (3.7%) | 15 (9.3%) | 13 (2.2%) | |
| 4 | 33 (4.4%) | 17 (10.6%) | 16 (2.7%) | |

AFP, alpha fetoprotein; BCLC, Barcelona clinic liver cancer; HBsAg, hepatitis B surface antigen; HCV, hepatitis C virus

nomogram shows the incidence of levels of probability of tumor recurrence within one year of LR. For example, a patient with HCC who underwent LR and showed a single tumor (0 point) 10.4 cm in size (44 points), tumor necrosis (47 points), MVI (96 points), grade 2 tumor differentiation (0 point), and an Ishak fibrosis score of 3 (i.e., noncirrhotic) (0 point) on pathological examination had an overall score of 187 and a probability of early recurrence of 0.31. Fig 2 shows the calibration plot created using bootstrapping. The x-axis shows the predicted probability of tumor recurrence within one year of LR, as evaluated by the nomogram, and the y-axis shows the observed rate of recurrence within one year. The observed probability of recurrence within one year was highly correlated with the nomogram's prediction. Therefore, the calibration plot showed high agreement between the nomogram's prediction and the actual observation.

The predictive power of the nomogram and that of the AJCC/UICC system [24] were compared using ROC curve analysis (Fig 3). The nomogram showed significantly higher discriminative accuracy in predicting one-year recurrence than the competing model: its c-index was 0.729 (95% CI, 0.683–0.775), which was substantially higher than that of AJCC/UICC (0.686; 95% CI, 0.643–0.728; $p$ = 0.004) (Fig 3). Bootstrap validation with 150 resamples resulted in a c-index of 0.722 for the nomogram.

## Discussion

In the present study, we developed a nomogram to predict the recurrence of HCC within one year of LR. The nomogram comprised six pathological variables: tumor size (> 5.0 cm vs. ≤ 5.0 cm); tumor number (multiple vs. solitary); MVI (yes vs. no); tumor necrosis (yes vs. no); cirrhosis (yes vs. no); and tumor differentiation (grade 3 or 4 vs. 1 or 2). It had a c-index of 0.722 and was shown to perform satisfactorily in predicting tumor recurrence within one year of LR. The nomogram was internally validated using bootstrapping. A strength of this study is that the variables used to construct the nomogram are well-known pathological features. All variables were dichotomous. The cutoff values of tumor size and number were based on the AJCC/UICC system [24]. Another advantage of the nomogram is its simplicity and ease of use. Furthermore, its predictive accuracy is superior to that of the AJCC/UICC system [24]. We adopted the 7[th] edition of the AJCC/UICC system [24], not the updated 8[th] edition [27].

**Table 2. The final model of the multivariate logistic regression analysis of recurrence of hepatocellular carcinoma within one year of liver resection.**

| Variables | Estimated Coefficient | OR | 95%CI | p |
|---|---|---|---|---|
| Tumor differentiation (Edmondson-Steiner grade [3–4 vs 1–2]) | 0.748 | 2.11 | 0.95, 4.59 | 0.062 |
| Tumor size (>5.0 vs ≦5.0cm) | 0.470 | 1.60 | 1.03, 2.48 | 0.036 |
| Tumor number (multiple vs single) | 1.056 | 2.88 | 1.81, 4.55 | <0.001 |
| Cirrhosis (yes vs no) | 0.596 | 1.81 | 1.23, 2.69 | 0.003 |
| Microvascular invasion (yes vs no) | 1.008 | 2.74 | 1.81, 4.22 | <0.001 |
| Tumor necrosis (yes vs no) | 0.497 | 1.64 | 1.11, 2.43 | 0.013 |

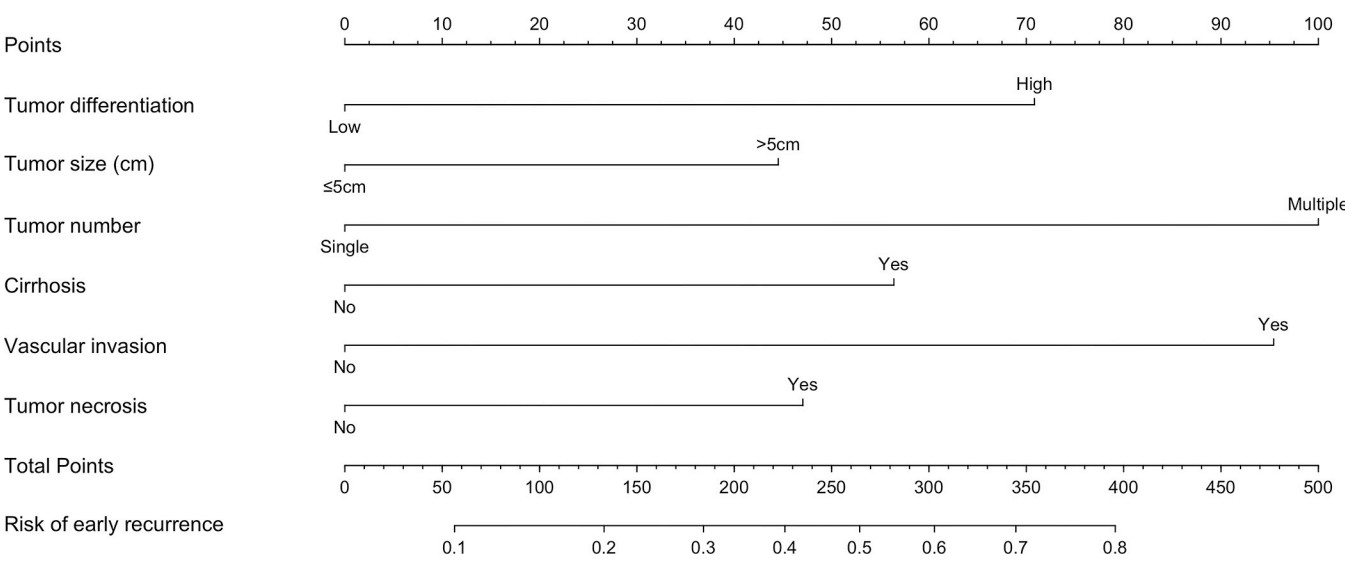

**Fig 1. A nomogram predicting recurrence of hepatocellular carcinoma within one year of liver resection.**

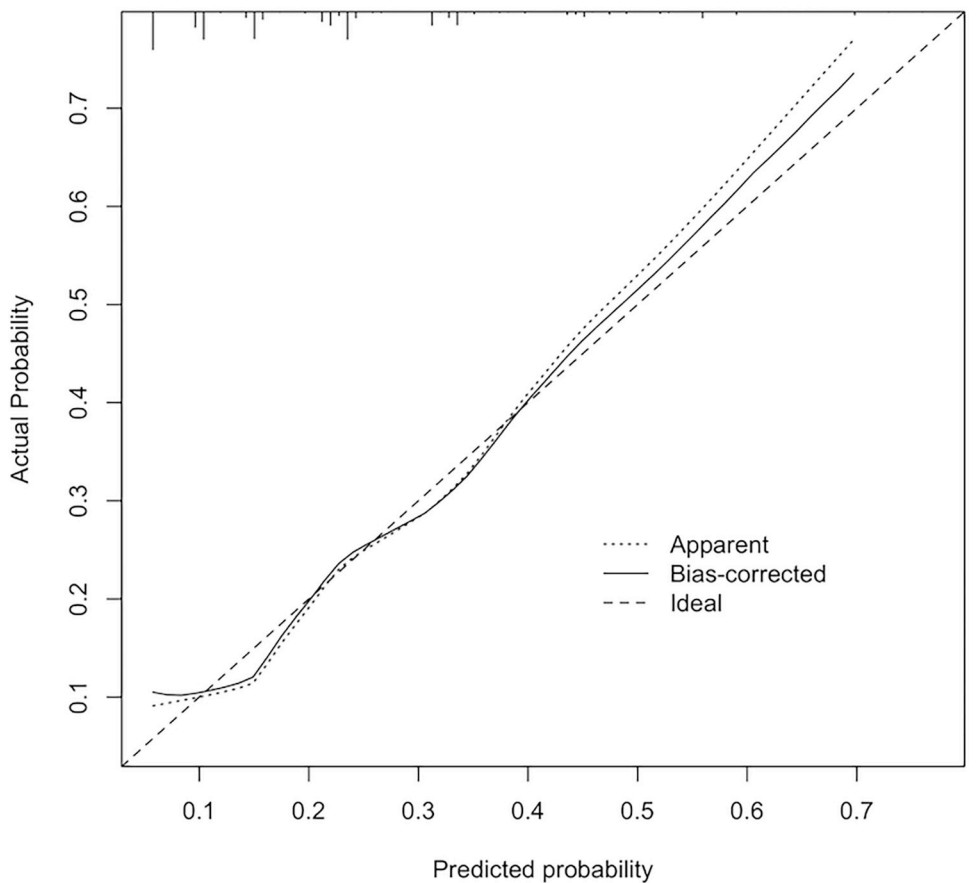

**Fig 2. The model's accuracy is visualized by comparing the predicted and actual probabilities of recurrence of hepatocellular carcinoma within one year of liver resection, showing the apparent predictive ability and bias due to overfitting.** The relative prevalence of probability levels is indicated by the vertical lines at the top of the plot.

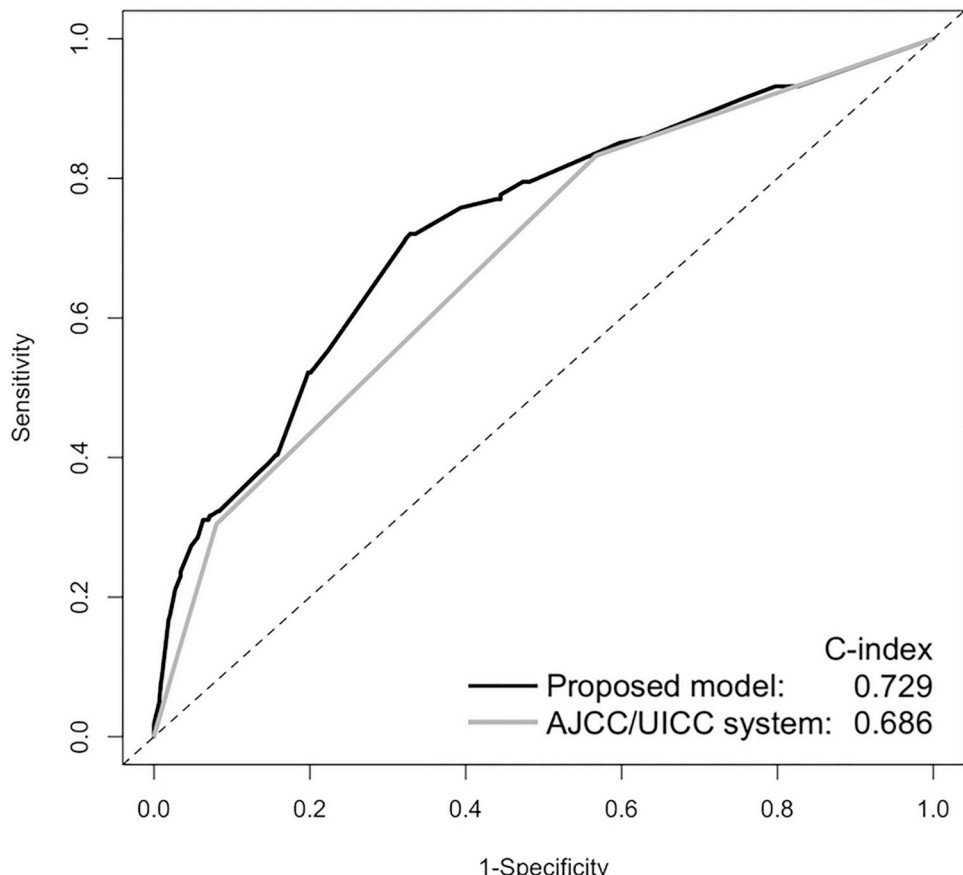

**Fig 3. The predictive power of the nomogram and that of the American Joint Committee on Cancer (AJCC)/ the Union for International Cancer Control (UICC) staging system were compared using receiver operating characteristic curve analysis.**

However, some studies have demonstrated that for patients with HCC undergoing surgery, the performance of the 8th edition is not superior to that of the 7th edition [28, 29]. The AJCC/UICC system [24] is commonly used to predict the prognosis of patients undergoing LR for HCC. However, its staging system does not take into account tumor differentiation, cirrhosis, and tumor necrosis. Therefore, its discriminative accuracy in predicting early recurrence is inferior to that of our nomogram.

To the best of our knowledge, only one previous study developed a model that included tumor necrosis as a variable to predict tumor recurrence in patients with HCC after LR [19]. Ling et al. enrolled 335 patients with HCC who underwent LR and had a solitary tumor of $\leq 3.0$ cm. A model with tumor necrosis, MVI, and tumor size could stratify the risk of recurrence and predict the probability of recurrence-free survival in this cohort ($p < 0.001$) [19]. However, whether this model could be extrapolated to patients with a tumor burden greater than a solitary HCC of $\leq 3.0$ cm is unclear.

Although nearly all HCC cells undergo hypoxia depending on their distance to blood vessels, tumor necrosis mainly occurs in large tumors due to severe and persistent hypoxia in the core area. The progression of an HCC is dependent on its microenvironment. Hypoxia and inflammation are two critical factors that affect the microenvironment of an HCC. Under hypoxia, the necrotic debris of tumor cells induce inflammation and changes in the microenvironment, resulting in immune evasion and ultimately, promotion of HCC metastasis [30].

In this study, the median OS was 2.274 years in patients with tumor recurrence within one year of LR, whereas it was more than 6 years in those without early recurrence. Therefore, we believe that patients with high risk of recurrence within one year of LR would be ideal candidates for adjuvant therapy. The STORM trial is the only international randomized controlled trial to evaluate the effect of adjuvant therapy following LR for HCC. Patients with HCC were randomized to sorafenib vs. placebo after curative treatment, which consisted of either LR (n = 900) or ablation therapies (n = 214). However, this trial failed to demonstrate any positive effects [31]. It assessed the risk of recurrence after LR based on pathological features and included only patients with an intermediate or high risk of recurrence. Patients undergoing LR were defined as having a high risk of recurrence if they had one tumor of any size plus MVI, satellite tumors, or poorly differentiated tumors or two or three tumors, each ≤ 3 cm in size. Intermediate risk was defined as a moderately or well-differentiated single tumor of ≥ 2 cm and the absence of MVI or satellite tumors. These criteria for the risk of recurrence were based on literature reviews [4, 32, 33]. However, we believe that this risk classification is not satisfactory for early recurrence. For example, a best-case scenario of intermediate risk could be a well-differentiated single tumor of 2.0 cm in a noncirrhotic liver and the absence of MVI, satellite tumors, and tumor necrosis. According to our nomogram, the total number of points for this scenario is 0 and the risk of early recurrence is below 0.1. A worst-case scenario of intermediate risk could be a moderately differentiated single tumor of 10.0 cm with tumor necrosis in a cirrhotic liver and the absence of MVI and satellite tumors. According to our nomogram, the total number of points is 147 and the risk of early recurrence is 0.22. Satellite tumors, included in the risk classification of the STORM trial, are a well-known pathological feature of HCC recurrence after LR [34]. Currently, there is no consensus on how to differentiate between multicentric and satellite tumors. Therefore, Zheng et al. [35] adopted multiple or satellite tumors as a risk factor to predict HCC recurrence beyond the Milan criteria after LR. In line with Zheng et al., this study classified satellite tumors as multiple tumors.

The failure of the STORM trial [31] may be due to the selection of wrong candidates and the wrong drug. Following the recent advances in systemic therapies, a phase 3 trial tested atezolizumab combined with bevacizumab versus sorafenib for unresectable HCC. It showed that the objective response, overall survival, and progression-free survival were better in the group receiving atezolizumab combined with bevacizumab than in the sorafenib group [36]. Furthermore, quality of life was also better in the atezolizumab combined with bevacizumab arm than in the sorafenib arm. Current guidelines do not recommend adjuvant therapy for HCC patients undergoing LR [37]. However, IMbrave050, a phase 3 study of the adjuvant atezolizumab + bevacizumab versus active surveillance of patients with HCC at high risk of disease recurrence following resection or ablation, showed positive results. Recurrence-free survival (RFS) was significantly improved with atezolizumab plus bevacizumab compared to active surveillance (HR = 0.72; 95% CI: 0.56–0.93; $p$ = 0.012) (https://doi.org/10.1158/1538-7445. AM2023-CT003). In line with other ongoing clinical trials of adjuvant therapy [38], we believe that adjuvant therapy might play an important role in the treatment of patients with HCC. Therefore, our nomogram could be integrated into clinical care in the future.

Early recurrence (i.e., recurrence within two years after surgery) is assumed due to progressive growth of microscopic residual tumors, whereas late recurrence is assumed due to de novo cancer development against the background of chronic liver disease [39]. Therefore, patients at high risk of early recurrence are potential candidates for adjuvant therapy [8]. The definitions in IMbrave050 of high risk of disease recurrence after resection are as follows: (1) ≤3 tumors, with the largest tumor >5 cm regardless of vascular invasion,[a] or poor tumor differentiation (Grade 3 or 4); (2) ≥4 tumors, with the largest tumor ≤5 cm regardless of vascular invasion,[a] or poor tumor differentiation (Grade 3 or 4); (3) ≤3 tumors, with the largest tumor ≤5 cm with

vascular invasion,[a] and/or poor tumor differentiation (Grade 3 or 4) ([a] microvascular invasion or minor macrovascular portal vein invasion of the portal vein—Vp1/Vp2) [40].

The strength of IMbrave050 is the simplicity of its criteria for high risk of HCC recurrence. However, there are limitations in the IMbrave050 criteria for high risk of HCC recurrence. Criterion 1 was ≤3 tumors, with the largest tumor >5 cm regardless of vascular invasion, or poor tumor differentiation (Grade 3 or 4). A single tumor >5.0 cm without MVI was classified as criterion 1; however, these patients are T1b according to the 8[th] edition of AJCC/TNM. According to our previous study, the prognosis of these patients is excellent. The 5-year OS rate was 82% for T1b tumors undergoing LR [41]. Criterion 2 was ≥4 tumors, with the largest tumor of ≤5 cm regardless of vascular invasion, or poor tumor differentiation (Grade 3 or 4). Of the patients undergoing resection in IMbrave050, 293 were in the atezolizumab plus bevacizumab arm; among these 293 patients, only 3 (1.0%) patients had ≥4 tumors. Of the 292 patients in the active surveillance arm, only 1 (0.3%) patient had ≥4 tumors. In our view, criterion 2 is not necessary and can be deleted.

There are some well-conducted studies on predicting HCC recurrence after LR. Shim et al. conducted a single center study that enrolled 1085 patients with mostly early-stage HCC undergoing LR. They randomly divided patients into derivation (n = 760) and validation (n = 325) cohorts. Variables including sex, albumin level, platelet count, MVI, and tumor volume were used to predict early recurrence, with a c-index of 0.69 for the derivation cohort and 0.66 for the validation cohort [11]. Although this study had a large sample size, it was limited by the lack of external validation of the model. Chan et al. enrolled 3903 patients with HCC undergoing LR from six centers around the world. The model to predict early recurrence was composed of the parameters sex, tumor size, tumor number, MVI, albumin-bilirubin (ALBI) grade, and serum AFP [8]. The c-index of the different cohorts ranged from 0.616 to 0.735. The strength of this study was its large sample size; it had also been externally validated.

In summary, compared to other studies [8, 11], the strength of our model is its simplicity. In addition, all the variables in our nomogram were from pathology and are mandatory in the pathology report. This minimized missing data and resulted in a robust model. The limitation of our model is that it was not externally validated. Furthermore, because this study was conducted in an HBV endemic area, its data may not be generalizable to other populations with different etiologies of chronic liver disease.

Novel histopathological examinations and genetic tests to predict HCC recurrence after LR may be more accurate compared to conventional models [42–45]. For example, macrotrabecular-massive HCC, which is a subtype identified from histopathological examination, has been found to be associated with poor RFS rates after LR [42]. Fibronectin is a matrix glycoprotein that promotes tumor progression [43]. Recent studies have shown that fibronectin expression level is associated with HCC recurrence after curative treatment [44, 45].

## Conclusion

We have developed a nomogram incorporating tumor necrosis to predict early tumor recurrence after curative resection in patients with HCC. The nomogram is simple and its predictive accuracy is satisfactory. The performance of the nomogram is superior to that of the 7[th] edition of the AJCC/UICC system. However, external validation is required to confirm the findings of this study.

## Supporting information

**S1 Raw data.**
(XLSX)

## Acknowledgments

The authors thank Cancer Center, Kaohsiung Chang Gung Memorial Hospital for the provision of HCC registry data. The authors thank Chih-Yun Lin and Nien-Tzu Hsu and the Biostatistics Center, Kaohsiung Chang Gung Memorial Hospital for statistics work. No conflict of interests.

## Author Contributions

**Conceptualization:** Yi-Hao Yen, Chih-Chi Wang.

**Data curation:** Chee-Chien Yong, Wei-Feng Li.

**Formal analysis:** Chih-Yun Lin.

**Supervision:** Hock-Liew Eng, Yueh-Wei Liu.

**Writing – original draft:** Fang-Ying Kuo.

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
