## [Decision Letter · Decision Letter 0]

9 Oct 2023

PONE-D-23-27028Tumor necrosis as a predictor of early tumor recurrence after resection in patients with hepatomaPLOS ONE

Dear Dr. Yen,

Thank you for submitting your manuscript to PLOS ONE. After careful consideration, we feel that it has merit but does not fully meet PLOS ONE’s publication criteria as it currently stands. Therefore, we invite you to submit a revised version of the manuscript that addresses the points raised during the review process.

We look forward to receiving your revised manuscript.

Kind regards,

Tyng-Yuan Jang

Academic Editor

PLOS ONE

Journal Requirements:

The authors thank Cancer Center, Kaohsiung Chang Gung Memorial Hospital for the provision of HCC registry data. The authors thank Chih-Yun Lin and Nien-Tzu Hsu and the Biostatistics Center, Kaohsiung Chang Gung Memorial Hospital for statistics work. This study was supported by Grant CMRPG8L0181 from the Chang Gung Memorial Hospital-Kaohsiung Medical Center, Taiwan. No conflict of interests.

4. Ethics statement appears in the Methods section of the manuscript AND at the end of the manuscript:

Your ethics statement should only appear in the Methods section of your manuscript. If your ethics statement is written in any section besides the Methods, please delete it from any other section. 

Reviewers' comments:

Reviewer's Responses to Questions

**Comments to the Author**

1. Is the manuscript technically sound, and do the data support the conclusions?

Reviewer #1: Yes

Reviewer #2: Yes

2. Has the statistical analysis been performed appropriately and rigorously? 

Reviewer #1: Yes

Reviewer #2: Yes

3. Have the authors made all data underlying the findings in their manuscript fully available?

Reviewer #1: Yes

Reviewer #2: Yes

4. Is the manuscript presented in an intelligible fashion and written in standard English?

Reviewer #1: Yes

Reviewer #2: Yes

5. Review Comments to the Author

Reviewer #1: Yen et al. enrolled a large cohort of patients with HCC undergoing resection. They used six predictive factors based on pathological features—tumor size > 5 cm, multiple tumors, high-grade tumor differentiation, tumor necrosis, microvascular invasion, and cirrhosis to predict early tumor recurrence. This model is internal validated and the performance of this model is superior to the AJCC stage, which is the commonest staging system used in the daily clinical practice. This study is well conducted; however, there are some issues need to be clarified.

1. In the discussion, the authors mentioned: However, a phase III trial of atezolizumab plus bevacizumab for high-risk HCC after curative resection is ongoing. Actually, the results of this trial had been released. Please update this information in your manuscript.

2. Please comments on the criteria for high risk of HCC recurrence in IMbrave050 trial, regarding on the strength and limitation of your model compared with the definition of high risk of HCC recurrence in IMbrave050 trial.

3. There are some well-conducted studies (Ann Surg 2015;261:939-46; Ann Surg Oncol 2022; 29:4291–4303; J Hepatol 2018;69:1284-1293; Surgery. 2018;163:1002-1007) for prediction of recurrence after curative liver resection for HCC. please comments on the strength and limitation of your model compared with these previous studies.

4. This study is conducted in a HBV endemic area. As such, data from the present study needs to be externally validated in Western patients (mostly with chronic HCV infection, and non-alcohol or alcohol steatohepatitis) to determine whether the results are generalizable to other patient populations. Please add in the limitations.

5. Please mention the further novel histopathological examinations and genetic tests on the prediction of recurrent tumors, which may be more accurate on the prediction of tumor recurrence after resection for HCC.

Reviewer #2: Yen et al. developed a nomogram including tumor necrosis as a factor to predict early tumor recurrence after resection. It might be a useful tool to discriminate patients who have high risk of recurrence and potentially benefit more from adjuvant systemic therapy. I just have some questions as follows:

1. The exclusion criteria of this study included neoadjuvant therapy. How about adjuvant therapy? Did any enrolled patient receive adjuvant therapy after surgery?

2. In your opinion, what is the recurrence rate that warrants adjuvant treatment?

3. IMbrave050 recently reported a positive result. Apart from the efficacy of atezo-bev, did their high-risk criteria meet your predictive model?

6. PLOS authors have the option to publish the peer review history of their article (what does this mean?). If published, this will include your full peer review and any attached files.

Reviewer #1: No

Reviewer #2: **Yes: **Po-Yao Hsu

---

## [Author Response · Author response to Decision Letter 0]

23 Oct 2023

5. Review Comments to the Author

Reviewer #1: Yen et al. enrolled a large cohort of patients with HCC undergoing resection. They used six predictive factors based on pathological features—tumor size > 5 cm, multiple tumors, high-grade tumor differentiation, tumor necrosis, microvascular invasion, and cirrhosis to predict early tumor recurrence. This model is internal validated and the performance of this model is superior to the AJCC stage, which is the commonest staging system used in the daily clinical practice. This study is well conducted; however, there are some issues need to be clarified.

1. In the discussion, the authors mentioned: However, a phase III trial of atezolizumab plus bevacizumab for high-risk HCC after curative resection is ongoing. Actually, the results of this trial had been released. Please update this information in your manuscript.

Response: Thank you for your comments. We have updated this information in the manuscript. Please see page 20, first paragraph.

2. Please comments on the criteria for high risk of HCC recurrence in IMbrave050 trial, regarding on the strength and limitation of your model compared with the definition of high risk of HCC recurrence in IMbrave050 trial.

Response: 

The definitions in IMbrave050 of high risk of disease recurrence after resection are as follows:

1. ≤3 tumors, with the largest tumor >5 cm regardless of vascular invasion,a or poor tumor differentiation (Grade 3 or 4);

2. ≥4 tumors, with the largest tumor ≤5 cm regardless of vascular invasion,a or poor tumor differentiation (Grade 3 or 4);

3. ≤3 tumors, with the largest tumor ≤5 cm with vascular invasion,a and/or poor tumor differentiation (Grade 3 or 4).

a Microvascular invasion or minor macrovascular portal vein invasion of the portal vein—Vp1/Vp2 [40]. 

The strength of IMbrave050 is the simplicity of its criteria for high risk of HCC recurrence. However, there are limitations in the Mbrave050 criteria for high risk of HCC recurrence. Criterion 1 was ≤3 tumors, with the largest tumor >5 cm regardless of vascular invasion, or poor tumor differentiation (Grade 3 or 4). A single tumor > 5.0 cm without MVI was classified as criterion 1; however, these patients are T1b according to the 8th edition of AJCC/TNM. According to our previous study, the prognosis of these patients was excellent. The 5-year OS rates was 82% for T1b tumors undergoing LR [40]. Criterion 2 was ≥4 tumors, with the largest tumor ≤5 cm regardless of vascular invasion, or poor tumor differentiation (Grade 3 or 4). Of the patients undergoing resection in IMbrave050, 293 were in the atezolizumab plus bevacizumab arm; among these 293 patients, only 3 (1.0%) had ≥4 tumors. Of the 292 patients in the active surveillance arm, only 1 (0.3%) had ≥4 tumors. In our view, criterion 2 is not necessary and can be deleted.

Please see pages 21 and 22. 

4. There are some well-conducted studies (Ann Surg 2015;261:939-46; Ann Surg Oncol 2022; 29:4291–4303; J Hepatol 2018;69:1284-1293; Surgery. 2018;163:1002-1007) for prediction of recurrence after curative liver resection for HCC. please comments on the strength and limitation of your model compared with these previous studies.

Response: 

Shim et al. conducted a single center study that enrolled 1085 patients with mostly early-stage HCC undergoing LR. They randomly divided patients into derivation (n = 760) and validation (n = 325) cohorts. Variables including sex, albumin level, platelet count, MVI, and tumor volume were used to predict early recurrence, with a c-index of 0.69 for the derivation cohort and 0.66 for the validation cohort [11]. Although this study had a large sample size, it was limited by the lack of external validation of the model. Chan et al. enrolled 3903 patients with HCC undergoing LR from six centers around the world. The model to predict early recurrence was composed of the parameters sex, tumor size, tumor number, MVI, albumin-bilirubin (ALBI) grade, and serum AFP [8]. The c-index of the different cohorts ranged from 0.616 to 0.735. The strength of this study was its large sample size; it had also been externally validated. 

In summary, compared to other studies [8, 11], the strength of our model is its simplicity. In addition, all the variables in our nomogram were from pathology and are mandatory in the pathology report. This minimized missing data and resulted in a robust model. The limitation of our model is that it was not externally validated. 

Please see pages 22 and 23.

5. This study is conducted in a HBV endemic area. As such, data from the present study needs to be externally validated in Western patients (mostly with chronic HCV infection, and non-alcohol or alcohol steatohepatitis) to determine whether the results are generalizable to other patient populations. Please add in the limitations.

Response: 

The limitation of our model is that it was not externally validated. Furthermore, because this study was conducted in an HBV endemic area, its data may not be generalizable to other populations with different etiologies of chronic liver disease.

Please see page 23, 2nd paragraph.

5. Please mention the further novel histopathological examinations and genetic tests on the prediction of recurrent tumors, which may be more accurate on the prediction of tumor recurrence after resection for HCC.

Response:

Novel histopathological examinations and genetic tests to predict HCC recurrence after LR may be more accurate than conventional models [41-44]. For example, macrotrabecular-massive HCC, which is a subtype identified from histopathological examination, has been found to be associated with poor RFS rates after LR [41]. Fibronectin is a matrix glycoprotein that promotes tumor progression [42]. Recent studies have shown that the fibronectin expression level is associated with HCC recurrence after curative treatment [43, 44]. 

Please see page 23.

Reviewer #2: Yen et al. developed a nomogram including tumor necrosis as a factor to predict early tumor recurrence after resection. It might be a useful tool to discriminate patients who have high risk of recurrence and potentially benefit more from adjuvant systemic therapy. I just have some questions as follows:

1. The exclusion criteria of this study included neoadjuvant therapy. How about adjuvant therapy? Did any enrolled patient receive adjuvant therapy after surgery?

Response: Thank you for your comments. None of the patients enrolled in this study underwent adjuvant therapy. Please see page 7, lines 2–3. 

2. In your opinion, what is the recurrence rate that warrants adjuvant treatment?

Response: It is difficult to define a cutoff value for the recurrence rate that warrants adjuvant treatment. However, early recurrence (i.e., recurrence within two years after surgery) is assumed due to progressive growth of microscopic residual tumors, whereas late recurrence is assumed due to de novo cancer development against the background of chronic liver disease [1]. Therefore, patients at high risk of early recurrence are potential candidates for adjuvant therapy [8]. Please see page 20, last paragraph and page 21, 1st paragraph. 

3. IMbrave050 recently reported a positive result. Apart from the efficacy of atezo-bev, did their high-risk criteria meet your predictive model?

Response: Their high-risk criteria did not meet our predictive model. 

The definitions in IMbrave050 of high risk of disease recurrence after resection are as follows:

1. ≤3 tumors, with the largest tumor >5 cm regardless of vascular invasion,a or poor tumor differentiation (Grade 3 or 4).

2. ≥4 tumors, with the largest tumor ≤5 cm regardless of vascular invasion,a or poor tumor differentiation (Grade 3 or 4).

3. ≤3 tumors, with the largest tumor ≤5 cm with vascular invasion,a and/or poor tumor differentiation (Grade 3 or 4).

a Microvascular invasion or minor macrovascular portal vein invasion of the portal vein—Vp1/Vp2 [40]. Please see page 21, first paragraph. 

In our model, six predictive factors, tumor grade (grade 3 or 4 vs. 1 or 2), tumor size (> 5.0 cm vs. ≤ 5.0 cm), tumor number (multiple vs. solitary), cirrhosis (yes vs. no), MVI (yes vs. no), and tumor necrosis (yes vs. no), were included in the final model for constructing the nomogram (Figure 1).

---

## [Decision Letter · Decision Letter 1]

3 Nov 2023

Tumor necrosis as a predictor of early tumor recurrence after resection in patients with hepatoma

PONE-D-23-27028R1

Dear Dr. Yi-Hao Yen,

We’re pleased to inform you that your manuscript has been judged scientifically suitable for publication and will be formally accepted for publication once it meets all outstanding technical requirements.

Kind regards,

Tyng-Yuan Jang

Academic Editor

PLOS ONE

Additional Editor Comments (optional):

Reviewers' comments:

Reviewer's Responses to Questions

**Comments to the Author**

1. If the authors have adequately addressed your comments raised in a previous round of review and you feel that this manuscript is now acceptable for publication, you may indicate that here to bypass the “Comments to the Author” section, enter your conflict of interest statement in the “Confidential to Editor” section, and submit your "Accept" recommendation.

Reviewer #1: All comments have been addressed

Reviewer #2: All comments have been addressed

2. Is the manuscript technically sound, and do the data support the conclusions?

Reviewer #1: Yes

Reviewer #2: Yes

3. Has the statistical analysis been performed appropriately and rigorously? 

Reviewer #1: Yes

Reviewer #2: Yes

4. Have the authors made all data underlying the findings in their manuscript fully available?

Reviewer #1: Yes

Reviewer #2: Yes

5. Is the manuscript presented in an intelligible fashion and written in standard English?

Reviewer #1: Yes

Reviewer #2: Yes

6. Review Comments to the Author

Reviewer #1: The author has fully answered all questions. We believe that through this nomogram including tumor necrosis can be to predict early tumor recurrence after resection. This would be a very useful tool to discriminate patients who have high risk of recurrence and potentially benefit more from adjuvant systemic therapy.

Reviewer #2: My comments have been addressed. Thus, I would like to recommend this manuscript for publication. Congratulations for your contribution.

7. PLOS authors have the option to publish the peer review history of their article (what does this mean?). If published, this will include your full peer review and any attached files.

Reviewer #1: No

Reviewer #2: **Yes: **Po-Yao Hsu

---

## [Editor Report · Acceptance letter]

8 Nov 2023

PONE-D-23-27028R1 

Tumor necrosis as a predictor of early tumor recurrence after resection in patients with hepatoma 

Dear Dr. Yen:

I'm pleased to inform you that your manuscript has been deemed suitable for publication in PLOS ONE. Congratulations! Your manuscript is now with our production department. 

Kind regards, 

on behalf of

Dr. Tyng-Yuan Jang 

Academic Editor

PLOS ONE